# Identification and Evaluation of (Non-)Intentionally Added Substances in Post-Consumer Recyclates and Their Toxicological Classification

Christian Rung [1], Frank Welle [1], Anita Gruner [1], Arielle Springer [1,*], Zacharias Steinmetz [2] and Katherine Munoz [2]

[1] Fraunhofer Institute for Process Engineering and Packaging IVV, Giggenhauser Straße 35, 85354 Freising, Germany
[2] Institute for Environmental Sciences (iES) Landau, University of Koblenz-Landau, Fortstraße 7, 76829 Landau, Germany
* Correspondence: arielle.springer@ivv.fraunhofer.de

**Abstract:** According to the European circular economy strategy, all plastic packaging placed on the market by 2030 has to be recyclable. However, for recycled plastics in direct contact with food, there are still major safety concerns because (non-)intentionally added substances can potentially migrate from recycled polymers into foodstuffs. Therefore, the European Food Safety Authority (EFSA) has derived very low migration limits (e.g., 0.1 µg/L for recycled polyethylene terephthalate (PET) and 0.06 µg/L for recycled high-density polyethylene (HDPE)) for recycled polymers. Thus, the use of recyclates from post-consumer waste materials in direct food contact is currently only possible for PET. A first step in assessing potential health hazards is, therefore, the identification and toxicological classification of detected substances. Within this study, samples of post-consumer recyclates from different packaging-relevant recycling materials (HDPE, LDPE, PE, PP, PET, and PS) were analyzed. The detected substances were identified and examined with a focus on their abundance, toxicity (Cramer classification), polarity (log $P$ values), chemical diversity, and origin (post-consumer substances vs. virgin base polymer substances). It was demonstrated that polyolefins contain more substances classified as toxic than PET, potentially due to their higher diffusivity. In addition, despite its low diffusivity compared to polyolefins, a high number of substances was found in PS. Further, post-consumer substances were found to be significantly more toxicologically concerning than virgin base polymer substances. Additionally, a correlation between high log $P$ values and a high Cramer classification was found. It was concluded that PET is currently the only polymer that complies with EFSA's requirements for a circular economy. However, better-structured collection systems and cleaning processes, as well as more analytical methods that enable a highly sensitive detection and identification of substances, might offer the possibility of implementing other polymers into recycling processes in the future.

**Keywords:** non-intentionally added substances (NIAS); food packaging; polymer contaminants; recycling; safety; exposure

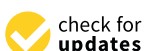



## 1. Introduction

Between the 1950s and 2015, global plastic production increased exponentially from 1.5 million to 322 million tons [1]. In Europe alone, over 61 million tons of new plastic products were manufactured in 2018. The largest share of produced plastics, with around 40%, is used for packaging materials [1,2]. The resulting plastic waste has an increasingly negative impact on both humans and the environment and has therefore become more relevant in recent years [3,4]. Therefore, the recycling of plastic waste is an essential factor in reducing plastic waste [5]. In 2018, on average, 42% of plastic waste produced in Europe was recycled with recycling rates ranging from 69% in Lithuania to 27% in France [6]. To improve this, the European Circular Economy Strategy was amended in 2018 and requires

all plastic packaging waste to be recyclable by 2030 [7,8]. The long-term goal is a European Circular Economy that promotes the reduction of waste and raw materials [9].

Especially for the food packaging industry, the fulfillment of this criterion is a great challenge as there are still major safety concerns for recycled plastics in direct contact with food. This is because the use of recycled food packaging increases not only potential sources of contamination but also the number and levels of chemicals that can migrate from the packaging into the food and thus potentially affect human health [10,11]. The safety of recycled plastics for food contact, therefore, depends on the migration rate of these non-intentionally added substances (NIAS) [12–15]. NIAS can originate from different sources, for example, post-consumer substances, contamination during recollection of the packaging materials, reaction products between different components in the packaging materials, degradation products of oligomers or approved additives during recycling. The use of antioxidant additives can significantly slow down the autoxidation of polyolefins. Therefore, their use as intentionally added substances (IAS) is necessary. Further, IAS can be components such as plasticizers, adhesives, inks, fillers and catalysts [16]. Commercially, amounts of 0.03 to 0.3 wt.% of antioxidant additives are used for the stabilization of polyolefins. However, the degradation products of these additives during oxidation are the main source of NIAS in polymers [17–23]. NIAS may also arise from impurities in the raw materials used for their production [24,25]. Another source of NIAS in recyclates for foodcontact is non-approved additives originating from cross-contamination of non-food packaging materials of the same polymer type which have been collectively recycled [14]. However, NIAS do not occur exclusively in recycled plastics but can also be found in virgin base polymer plastic products [16]. Since most of these substances are only loosely bound to the polymer via non-covalently interactions, i.e., van der Waals forces, they can migrate into liquids and solids or be released through volatilization at all stages of the plastics' life cycle [26]. When analyzing recyclates, it is important to distinguish between substances that originate from virgin base polymer packaging materials and post-consumer materials. For this study, virgin base polymer substances, such as additives, are defined as substances that occur in unused and raw materials which have never been exposed to any processing. Post-consumer substances, on the other hand, are defined as substances from a product that has completed their intended use and has been discarded for disposal, having completed its use as a consumer item.

NIAS can be determined in post-consumer recyclates and should be monitored. In addition, substances should be evaluated regarding their toxicological profile. Without knowledge about the chemical identity of potential contaminants and NIAS, any contaminant must, according to the European Food Safety Authority (EFSA), be considered in a worst-case scenario as a chemical with toxic effects and as a potential DNA-reactive carcinogen [12]. Following this conservative assumption, EFSA has derived very low migration limits for post-consumer recyclates (e.g., 0.1 μg/L for recycled polyethylene terephthalate (PET) and 0.06 μg/L for recycled high-density polyethylene (HDPE)) [27]. As a result, the use of recyclates from post-consumer waste for packaging materials in direct food contact is currently not possible for high-diffusive polyolefins such as HDPE. This prevents a closed circular economy for the most commonly used plastic packaging materials. Reliable identification and toxicological classification of contaminants and other NIAS are, therefore, of great importance. Currently, plastic recycling in food packaging is limited to PET recyclates. Recycled PET is characterized by its low diffusivity and inert nature. Therefore migration of substances into foodstuffs is relatively low [12–14]. Other common polymers, such as high-diffusive low-density polyethylene (LDPE) and polypropylene (PP), and less-diffusive polystyrene (PS), still need further research. Polyolefins are high-diffusive polymers; therefore, recyclates of these polymers often show increased concentrations of contaminants and NIAS [28]. In addition, due to their lipophilic character, they absorb and adsorb substances with similar physicochemical properties [29]. Thus, the polarity, expressed as the octanol-water partition coefficient (log $P$), is a suitable parameter to determine sorption and desorption kinetics in polymers [30–33].

One of the most common methods to classify identified substances based on their toxicity is the Cramer classification and the underlying "Threshold of Toxicological Concern" (TTC) concept [34,35]. This approach can be used when the chemical structure of a substance is known, but there is limited substance-specific toxicity data [36]. Here, a decision tree is applied to estimate the toxicological effect of a substance on the basis of its chemical structure. Each Cramer class is linked to a specific human exposure level (oral intake) below which substances are expected to pose an insignificant risk to human health [37,38]. Moreover, the TTC concept can be used to estimate potential genotoxic (gtc) or non-genotoxic carcinogenic (ngtc) effects.

To the authors' knowledge, research studies systematically considering the diversity of NIAS in post-consumer recyclates and their toxicological classification on a large sample scale are still absent. Therefore, there is a need to elaborate on this aspect with a comparable and reproducible experimental approach. As recycling in the food packaging sector is currently not possible for polyolefins and PS and migration limits have only been derived for HDPE, the aim of the present study was the screening of recyclates in terms of their chemical diversity and toxicological classification [39,40]. For this purpose, a total of 179 post-consumer recyclate samples of polymers characterized as HDPE, LDPE, PE, PS, PP and PET were collected from different European countries and analyzed to identify and characterize the absorbed chemicals. Samples were classified as PE when no further classification to HDPE or LDPE was possible. A risk assessment based on the chemical nature of these substances and their toxicological classification was performed using in silico models, namely the Cramer classification and ISS model (carcinogen vs. non-carcinogen).

Based on the low diffusivity of PET, we (1) expected a lower number of substances classified as toxic in PET recyclates than in polyolefin and PS recyclates. Further, we (2) expected post-consumer substances to be toxicologically more concerning than substances from virgin base polymer materials in in silico tests. Finally, due to the high bioaccumulation potential of lipophilic substances we (3) assumed a relationship between a high partitioning coefficient (log *P*) of the substances and their toxicological classification.

## 2. Results

### 2.1. Sample Screening for Post-Consumer Substances

2.1.1. Extract Screening

Analysis to identify and characterize the absorbed chemicals via headspace gas chromatography with flame ionization detection (GC-FID) and headspace gas chromatography with mass spectrometry (GC-MS) showed that in 179 analyzed sample extracts, a total of 205 different substances were found. Out of these, 175 substances were identified while 30 substances did not result in a clear match with the database and were thus marked as unknown. It is important to note that unknown substances were only listed if detected more than twice.

Considering the results of the analyzed recyclates (physicochemical properties and toxicological classification for each polymer type), the number of detected substances differed between polymers. Around 47% of the detected substances were found in PP, while the largest proportion of the total number of detected substances out of all samples, 25%, was found in HDPE. Also, the average number of substances found per sample differed within the respective polymers. While on average, only 7.8 substances per sample were found in PET, an average of 20 substances per sample were detected in PS (Figure 1). A comparison between the number of detected substances in each polymer showed a significant difference ($p < 0.001$ ***, df = 5, ANOVA). In addition, the proportion of unknown substances was lowest for LDPE at 1.7% and highest for PET at 37.5%.

In order to evaluate the chemical diversity of the respective post-consumer recyclates, diversity indices (Shannon and Simpson) were applied [41,42]. Further evenness was used as a measure of unequal distribution [43]. The calculated diversity indices (Table 1) showed that the substances in PET were chemically less diverse (lowest Shannon and Simpsonindices) and more unevenly (low evenness) distributed compared to other polymers.

This indicates that PET samples were more likely to contain the same substances. Further, a comparison between all polymers for the Shannon and Simpson diversity indices and the evenness showed significant differences throughout (Shannon: $p \leq 0.001$ ***, KS-test; Simpson: $p = 0.001$ ***, KS-test; evenness: $p = 0.009$ **, KS-test).

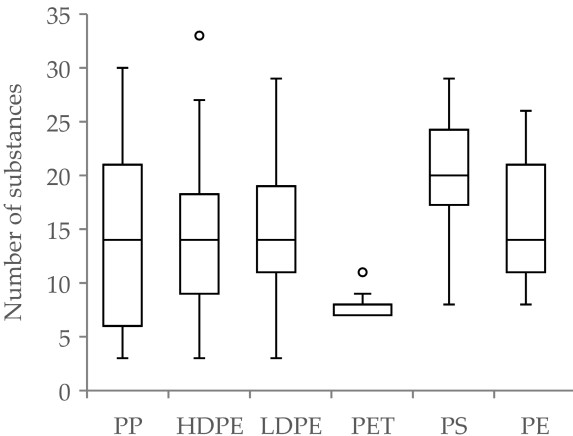

**Figure 1.** Boxplot analysis for the number of detected substances for each polymer. This was done by counting the number of detected substances in each sample. (° = Outlier).

**Table 1.** Calculated diversity indices (Shannon-index; Simpson-index) and evenness for each polymer.

|  | **HDPE** | **LDPE** | **PE** | **PET** | **PP** | **PS** |
|---|---|---|---|---|---|---|
| Shannon index | 3.20 | 3.67 | 2.92 | 0.90 | 4.02 | 3.41 |
| Simpson index | 0.95 | 0.97 | 0.98 | 0.56 | 0.97 | 0.96 |
| Evenness | 0.82 | 0.90 | 0.76 | 0.62 | 0.88 | 0.91 |

### 2.1.2. Comparison between Virgin Base Polymer Substances and Post-Consumer Substances

Classification of the substances according to their origin (virgin base polymer vs. post-consumer polymer) showed that the share of substances from virgin base polymers was lowest in PET at 12.5% and highest in PE at 62%. Further, for all polymers, the number of substances from virgin base polymers and the number of substances from post-consumer materials were compared in terms of their toxicological classification. It was shown that post-consumer substances were more often classified as toxic than virgin base polymer substances. For the parameters, Cramer classes higher than class I (Cramer class > I) and ngtc alerts significant differences were found while no significant difference regarding the number of ngtc alerts was observed (Cramer: $p \leq 0.001$ ***, df = 1, ANOVA; gtc: $p = 0.491$, df = 1, ANOVA; ngtc: $p \leq 0.001$ ***, df = 1, ANOVA).

### 2.1.3. Analysis of the Partitioning Coefficient

For the substances, a range of log $P$ values from 1.55 to 10.05 was determined while the log $P$ values of the polymers themselves range approximately from 1.05 to 2.78. In regard to our classification system, no substances were assigned to class 1, 9 substances to class 2, 25 substances to class 3 and 141 substances to class 4. For 8 substances, it was not possible to determine a log $P$ value due to a lack of structural information. Considering the number of substances found in the polymers that belong to log $P$ class 4, most substances were found in PP, while only three were found in PET (Figure 2a). A significant relationship between the toxicity and log $P$ was demonstrated for the parameters Cramer class > I and ngtc alerts but not for the parameter gtc alerts (Cramer: $p \leq 0.001$ ***, df = 1, ANOVA; gtc: $p = 0.099$, df = 1, ANOVA; ngtc: $p = 0.03$ *, df = 1, ANOVA).

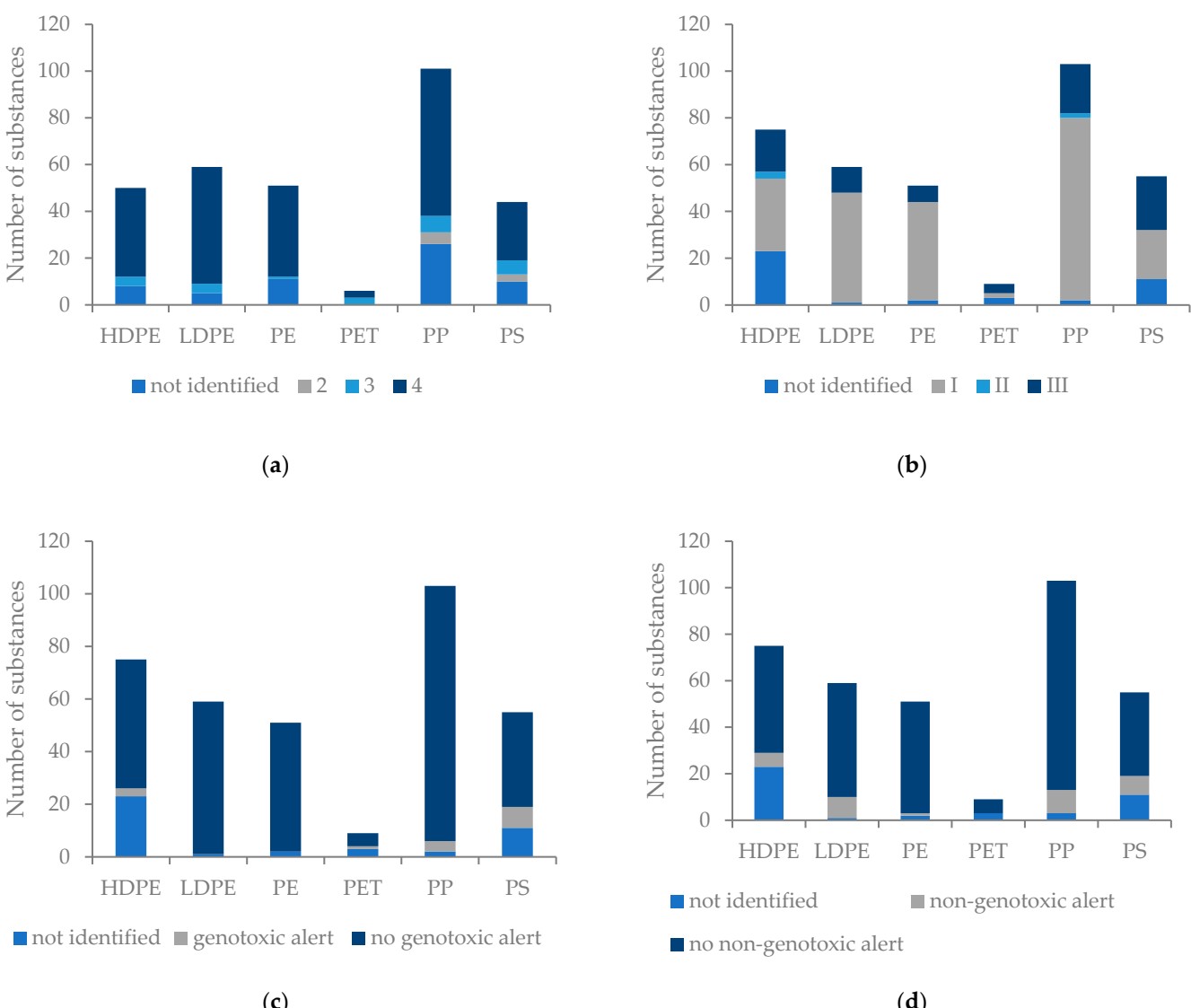

**Figure 2.** Bar plots with (**a**) the no. of substances from each log *P* class for each polymer, class 1: log *P* = 0–1, class 2: log *P* = 1–2, class 3: log *P* = 2–4, class 4: log *P* ≥ 4; (**b**) the no. of substances for each Cramer class; (**c**) no. of substances with gtc alerts; (**d**) no. of substances with ngtc alerts.

### 2.1.4. Analysis of the Toxicological Classifications

Out of the identified substances, 51 substances had a Cramer class > I, while 8 and 17 substances were found to have a potential gtc and ngtc alert. Classification of the substances according to toxicological parameters showed that PS had the highest number of substances with Cramer class > I assignments. PET showed the smallest number of substances for Cramer class > I, gtc alerts and ngtc alerts (Figure 2b–d). A comparison of the toxicological classification of the polymers showed that there was a significant difference between the number of substances with Cramer class > I and gtc alerts but no significant difference regarding the number of ngtc alerts (Cramer: *p* ≤ 0.001 ***, df = 5, ANOVA; gtc: *p* = 0.002 **, df = 5, ANOVA; ngtc: *p* = 0.18, df = 5, ANOVA).

### 2.1.5. Comparison of the Polymers with PET

PET was compared with the polyolefins and PS regarding their toxicity, polarity, and number of virgin base polymer substances (Table 2). It was shown that substances in all polymers, but PS differed significantly from PET in terms of their log *P* values. In addition, Cramer classes > I and gtc alerts showed a significant difference for LDPE, PE, and PP. Thus,

for these parameters, PET was shown to have a higher percentage of as toxic classified substances. For the parameter virgin base polymer substances and ngtc alerts, no significant difference was found for any polymer compared to PET.

**Table 2.** Results of the Dunnett's tests for each polymer with PET for the parameters no. of substances with Cramer classification > I and no. of substances with log $P$ class 4 and results of the $X^2$-tests for the parameters no. of substances with gtc and ngtc alerts and no. of virgin base polymer substances. Note: * significance with $p \leq 0.05$; ** significance with $p \leq 0.01$; *** significance with $p \leq 0.001$.

| | *p*-Values | | | | |
|---|---|---|---|---|---|
| | **HDPE** | **LDPE** | **PE** | **PP** | **PS** |
| Cramer class > I | 0.404 | 0.002 ** | 0.002 ** | 0.003 ** | 0.580 |
| gtc alert | 1.000 | <0.001 *** | 0.006 ** | 0.002 ** | 1.000 |
| ngtc alert | 1.000 | 1.000 | 0.213 | 0.530 | 1.000 |
| Virgin base polymer substances | 1.000 | 1.000 | 0.840 | 1.000 | 1.000 |
| Log $P$ | 0.009 ** | 0.002 ** | 0.001 *** | 0.017 * | 0.777 |

### 2.2. Origin of the Most Frequently Detected Substances

The most frequently found substances were virgin base polymer substances and were introduced to polymers either as additives from the production processes or as degradation products of the polymers. The most frequently found substance was oxidized Irgafos 168. This substance results from the degradation of Irgafos 168, which is a common organophosphite processing stabilizer additive. It was detected in all polymers except PS and PET. In addition, a large number of alkanes was identified in all polymers except PS and PET. These originate from the base polymers themselves. The post-consumer material substances originate from packaging contents. These substances are mostly food-grade additives and cosmetic products. However, it is also possible that they originate from the cleaning process of the recyclates (Table 3). Within Table 3 also, the origin of the substances was described as virgin base polymer substances (VBP) and post-consumer material substances (PCM). However, this classification is not always clear-cut because some substances may have several contamination pathways. Post-consumer mixed plastics might be cross-contaminated during recollection and recycling with packed content from other packages or with additive and their degradation products from other polymer types. In addition, also a clear classification of the packed content for example in food, cosmetic, household cleaners and technical products is not possible. Scientific literature with a clear classification is to the knowledge of the authors not available, and the classification is therefore based on experiences in our group.

### 2.3. Quantification of Additives

The quantification of additives Irgafos 168, oxidized Irgafos 168, Irganox 245, Irganox 1010, Irganox 1076 and Irganox 1330 in different types of polymers is shown in Figure 3a–f. It was shown that the distribution of the substances varies strongly within a polymer class. A significant difference in the quantified concentrations of the substances between different polymers was shown in all additives (Irgafos 168: $p \leq 0.001$ ***, df = 6, log-Tukey; oxidized Irgafos 168: $p \leq 0.001$ ***, df = 6, log-Tukey; Irganox 245: $p \leq 0.001$ ***, df = 6, log-Tukey; Irganox 1010: $p \leq 0.001$ ***, df = 6, log-Tukey; Irganox 1076: $p \leq 0.001$ ***, df = 6, log-Tukey; Irganox 1330: $p \leq 0.001$ ***, df = 6, log-Tukey).

### 2.4. Headspace Quantification of Limonene

Limonene, which is a post-consumer volatile compound and is found in nearly all investigated post-consumer recyclate samples, was quantified as a key substance for the volatile compounds [44]. It is introduced into packaging by fragrances, for example, cosmetics and foodstuffs [45]. Especially in the case of highly diffusive polymers such as PP and HDPE, very high concentrations were also found in some of the outliers (Figure 4).

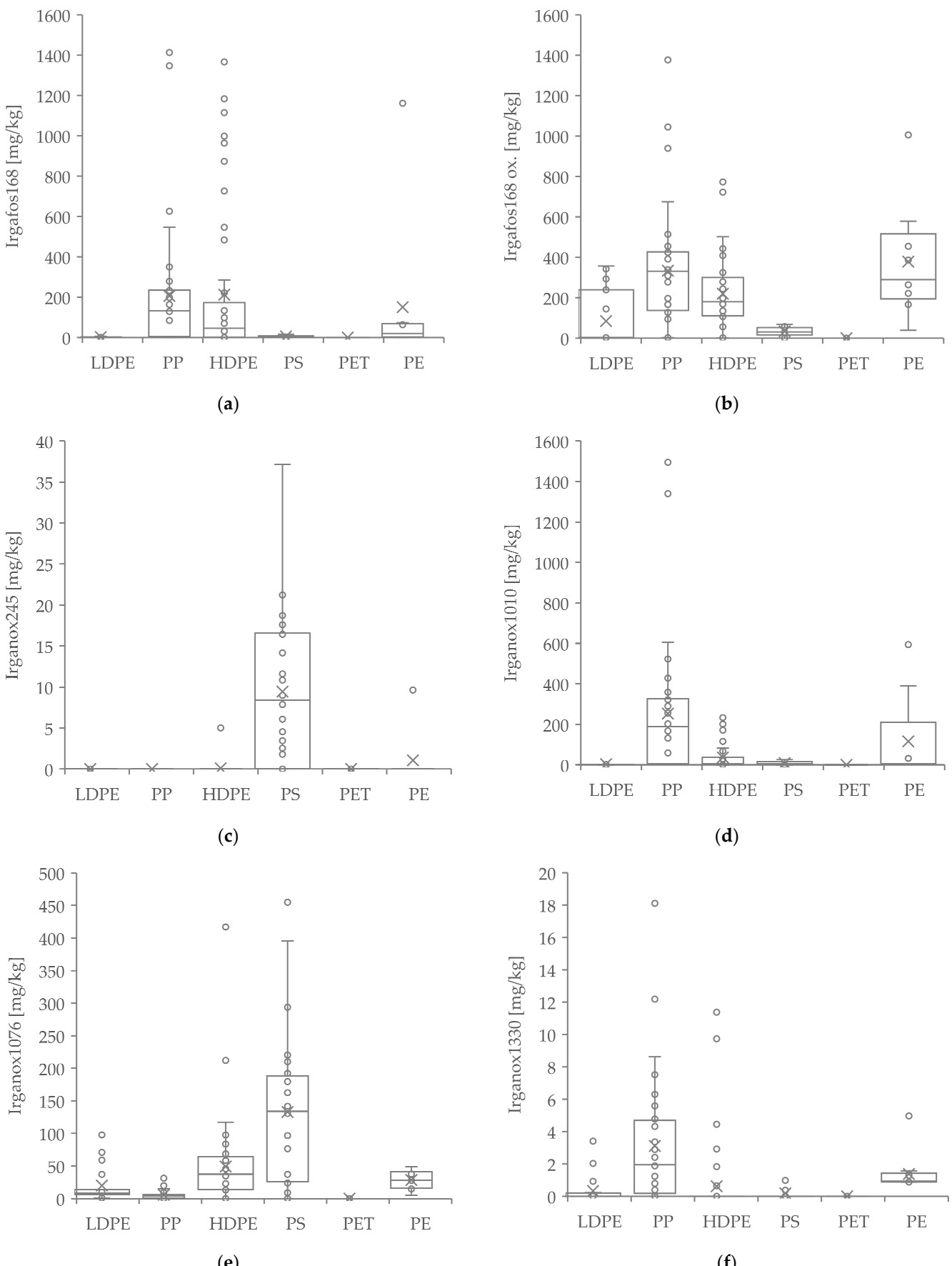

**Figure 3.** Quantification of additives Irgafos 168; oxidized Irgafos 168; Irganox 245; Irganox 1010; Irganox 1076 and Irganox 1330 (**a–f**) in different types of polymers. (× = Mean, ° = Outlier).

**Table 3.** Most frequently detected substances with information about the respective polymers they were detected in, toxicological parameters (Cramer classification, gtc and ngtc alerts), possible origin, and type of substances with virgin base polymer substances (VBP) and post-consumer material substances (PCM).

| Substance (mol. Weight [g/mol])-CAS No. | Polymer | Cramer Class | Gtc Alerts | Ngtc Alerts | Possible Type of Substance | Possible Origin |
|---|---|---|---|---|---|---|
| Oxidized Irgafos 168 (662)-95906-11-9 | HDPE, LDPE, PE, PP, PS | III | - | - | VBP | Additive |
| Irgafos 168 (646)-31570-04-4 | HDPE, LDPE, PE, PP | III | - | - | VBP | Additive |
| n-Tetracosane (338)-646-31-1 | HDPE, LDPE, PE, PP | I | - | - | VBP | Polymer |
| n-Octadecane (254)-593-45-3 | HDPE, LDPE, PE, PP | I | - | - | VBP | Polymer |
| n-Eicosane (282)-112-95-8 | HDPE, LDPE, PE, PP | I | - | - | VBP | Polymer |
| n-Docosane (310)-629-97-0 | HDPE, LDPE, PE, PP | I | - | - | VBP | Polymer |
| n-Hexadecane (226)-544-76-3 | HDPE, LDPE, PE, PP | I | - | - | VBP | Polymer |
| n-Tetradecane (198)-629-59-4 | HDPE, LDPE, PE, PP | I | - | - | VBP | Polymer |
| n-Hexadecanoic acid (256)-57-10-3 | LDPE, PP, PS | I | - | - | PCM | Packed content |
| Oleic Acid (282)-112-80-1 | LDPE, PE, PP, PS | I | - | - | PCM | Packed content |
| n-Hexacosane (366)-630-01-3 | HDPE, LDPE, PE, PP | I | - | - | VBP | Polymer |
| Di-(ethylhexyl)-isophthalat (390)-137-89-3 | LDPE, PP, PS | I | - | + | PCM | Packed content |
| Octadecanoic acid (284)-57-11-4 | LDPE, PP, PS | I | - | - | PCM | Packed content |
| 1-Docosene (308)-1599-67-3 | HDPE, LDPE, PE, PP | I | - | - | VBP | Polymer |
| 1-Octadecene (252)-112-88-9 | HDPE, LDPE, PE | I | - | - | VBP | Polymer |
| 3-Eicosene, (E) (280)-74685-33-9 | HDPE, LDPE, PE | I | - | - | PCM | Packed content |
| n-Pentacosane (352)-629-99-2 | HDPE, LDPE, PE, PP | I | - | - | PCM | Packed content |
| 1,3-Diphenylpropane (196)-1081-75-0 | PS | III | - | - | PCM | Packed content |
| 1,1′-(2-methyl-2-(phenythio) cyclopropylidene) bis-benzene (316)-56728-02-0 | PP, PS | III | - | - | PCM | Packed content |
| n-Tetracosene (336)-10192-32-2 | HDPE, LDPE | I | - | - | VBP | Polymer |
| trans-(2,3-Diphenylcyclopropyl) methyl phenylsulfoxide (332)-131758-71-9 | PE, PP, PS | III | - | - | PCM | Packed content |
| Isopropyl Palmitate (298)-142-91-6 | HDPE, LDPE, PE, PP | I | - | - | PCM | Packed content |
| 3-(1-(4-Cyano-1,2,3,4-tetrahydronaphthyl)) propanenitrile (210)-57964-40-6 | PP, PS | III | - | - | PCM | Packed content |
| 2-(1-(4-Cyano-1,2,3,4-tetrahydronaphthyl)) propanenitrile (210)-57964-39-3 | PP, PS | III | - | - | PCM | Packed content |
| (2,2) Paracyclophane (208)-1633-22-3 | PS | III | - | - | PCM | Packed content |
| Isopropyl Myristate (270)-110-27-0 | HDPE, LDPE, PE, PP | I | - | - | PCM | Packed content |

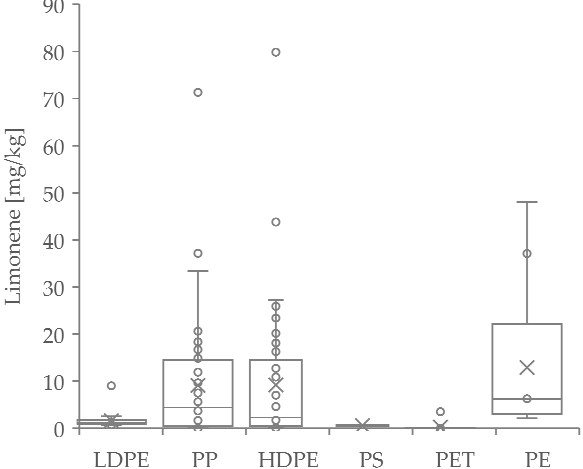

**Figure 4.** Boxplot analysis of limonene concentrations in the analyzed polymers. (× = Mean, ° = Outlier).

## 3. Discussion

### 3.1. Chemical Differences between Polymers and Their Substances

A comparison of the polymers showed that the number of detected substances differed between polymers. Furthermore, the chemical diversity, based on the Shannon and Simpson indices and evenness, showed a significant difference between the polymers (Table 3). This means that the substances in each polymer were differently distributed, which shows how strongly the polymers differ in their chemical and toxicological properties. In the case of PET, values for each diversity index were slightly lower compared to all other polymers. This could indicate a trend in PET for a lower diversity, meaning that fewer different substances were found. The evenness is also lower in PET, indicating an unequal distribution. This is due to the fact that among the few substances found in PET, a cyclic dimer of PET which occurs inherently in PET, was found proportionally more often than other substances (in 20 of 25 samples). In the other polymers, though more substances were found, they did not differ as much in their abundance.

The comparison of polyolefins and PS with PET showed that the proportion of substances with gtc alerts and substances with Cramer class > I was higher in numbers in PET in comparison to LDPE, PE, and PP. This high proportion in PET can be explained by the fact that a total of just eight substances were detected, six of which were assigned to Cramer class > I. In the other polymers, considerably more substances were found, which makes the proportion of toxic substances lower. It is, therefore, important to consider how many toxic classified substances were found in total in a polymer. This shows that overall fewer substances from Cramer class > I and substances with gtc alert were found in PET. This is also evident when looking at the average number of substances found in a sample. On average, only 7.8 substances were found in PET, while an average of 13 to 16 substances were detected in polyolefins and 20 in PS per sample. It can therefore be concluded that the proportion of toxic classified substances in PET is higher compared to the other polymers, but the total number of as toxic classified substances is considerably lower. Consequently, recycled PET contains the lowest number of toxicologically concerning substances. This is due to the low diffusivity and the inert character of PET [17]. Because of their high diffusivity, polyolefins absorb a high amount of contaminants they have been in contact with, whereas in the case of PET, the contaminants are generally absorbed or adsorbed in a smaller amount into the polymer matrix due to the slower sorption process. As a result, the migration rates of substances from the polymers into packaging content materials are higher with polyolefins than with PET due to different polymer properties [12,28,46].

However, the determination of specific values for sorption or migration rates is complex and only possible to a limited extent because these rates do strongly depend on the characteristics of the substances. In addition, many of the factors affecting migration can only be estimated. Factors such as storage temperature, properties of the packed contents, storage time and thickness of the packaging materials and how strongly the substance is bound into the matrix are critical when calculating migration rates [47,48]. Currently, the two most accurate and most applied prediction models for calculating migration rates are using the Piringer or Welle equation [49]. The Piringer model is based on the molecular weight of the migrating substance and the average molecular weight of the polymer [50], while the Welle equation is based on the molecule volume rather than the molecular weight [51]. However, it has been shown that in PET, low molecular weight substances have higher diffusion rates than high molecular weight substances [52,53]. This is because higher molecular substances have much higher activation energies for diffusion [54]. A thorough assessment of the toxicity is not possible without knowledge about the possible migration rates of the respective substances since detected substances only pose a risk if they can migrate from the polymer matrices into food.

### 3.2. Origin of the Most Frequently Detected Substances, Additives and Limonene

Alkanes are mostly found in polymers as residual monomers, which are also formed during the polymerization process [55]. The migration behavior of alkanes is strongly

dependent on the chain length, meaning the molecular weight [56]. No experimentally obtained data on potential modes of action on human health and the environment were found for any of the alkanes. The oligomers and residual monomers are, therefore, NIAS that are nevertheless known [15].

To improve the properties of polymers and extend their longevity, various substances are being added to polymers. These additives are, among others, plasticizers, thermal stabilizers, flame retardants, antioxidants, and light and heat stabilizers [47]. Irgafos 168, which was identified in the polyolefins, is a widely used processing stabilizer and antioxidant for polymers [47,57]. Oxidized Irgafos 168 is one of the main degradation products of Irgafos 168 and can be produced by extrusion, storage and solar radiation [57]. In this study, oxidized Irgafos 168 was detected in all polymers except PET. Investigations on the hazard potentials of Irgafos 168 and its degradation products for human health showed that Irgafos 168 poses little to no risk due to low toxicity as well as a low migration rate [57]. Oxidized Irgafos 168 is expected to be more toxic [58], but here too, the migration rates are low, which means that a hazard potential can be dismissed [57,59]. The EU has assigned a specific migration limit for Irgafos 168 but not for oxidized Irgafos 168 [60].

It was shown that about half of the most frequently found substances were either deliberately added as additives or were formed as known NIAS during polymerization. None of these substances are considered to pose a health risk. However, it is important to investigate possible degradation products as they may have higher migration rates and higher toxicological potential. The example of Irgafos 168 showed that a specific migration limit was set by the EU, but not for its degradation products. This means that a reliable safety classification can only be guaranteed to a limited extent. Looking at the collected data, it also appears that oxidized Irgafos 168 was detected more often than Irgafos 168, which shows that migration limits for degradation products need to be included in standard protocols for migration monitoring [61]. The other half of the most frequently detected substances were post-consumer material substances originating from packed contents or processing of the recyclates. A small portion of these substances was classified as Cramer class III substances and/or had (n)gtc alerts and, therefore, might pose health risks. While concerning substances were found in the oligomers and PS, none were found in PET. This underlines the potential safety risk of the potential inclusion of oligomers and PS into a circular food packaging recycling stream.

The post-consumer-substance limonene is also used, besides its application in cosmetics and household products, as a major flavor and fragrance additive in food items, such as fruit juices and ice cream [45]. As a food ingredient, it is not under any restriction [62]. As an ingredient for cosmetic products, it must be declared in the International Nomenclature of Cosmetic Ingredients (INCI) list due to allergen labeling for concentrations above 0.001% for leave-on products and 0.01% for rinse-off products [63]. It is almost completely absorbed via the digestive tract and inhalation and has an irritant and sensitizing/allergenic effect on mucous membranes and skin. Autoxidation with air and light further enhances the effect. There is currently insufficient evidence to exclude cancerogenicity, reproductive toxicity and mutagenicity [64]. However, it should be noted that toxicity has so far only been described at significantly higher concentrations. Nevertheless, not only is acute and chronic toxicity a factor with limonene, but also human perception. Since the substance can be perceived ortho- and retronasally, it could transport fruity, citrus and fresh off-flavors from the packaging into the packaged medium. Its oxidation products, such as carvone and carveol, can also release aromas of terpene, mint or caraway. For (R)-(+)-limonene, the odor threshold in water is very low at 10 µg/kg, and the taste threshold is 210 µg/kg. Carvone has an odor threshold in water at about 530 µg/kg [65–67]. Assuming that a 500 mL bottle of water would be produced from the investigated materials (approx. 35 g), the perception threshold of 10 µg/kg would be reached if there were 0.14 mg/kg limonene in the packaging material. As can be seen from Figure 4, the mean value of all polyolefins is clearly above this value. Therefore, the analyzed polyolefin samples would be sensory unsuitable for food contact if total migration occurs. For the quantification of additives, it

is clearly visible that the distribution of the substances sometimes varied strongly within a polymer class. We could show significant differences between polymers for all additives. The comparison between Irgafos 168 (Figure 3a) and its oxidized degradation product (Figure 3b) is of interest, as the concentrations of the oxidized substance are significantly higher than those of the non-oxidized additive, especially in LDPE, PP, HDPE, and PE. Although general limits for the use of additives are not defined, the high amount of oxidized Irgafos 168 suggests an oxidative degradation of the material, which could also lead to other oxidation products for which limits could be defined and possibly exceeded.

### 3.3. Recycling Potential and Recyclate Preparation

Since the diffusion behavior of PS is low, although not as low as that of PET, it was expected that the number of substances found in PS would be between that of PET and the polyolefins [28]. Therefore, it could have been expected that mainly PS-typical substances, such as PS dimers, would be found. Instead, we found a high number of detected substances in PS. This might be due to the way the samples were collected. The extent to which packaging materials can be contaminated depends strongly on the type of collection system and how the waste is being processed. In mixed recollection schemes, the post-consumer polymers have contact with lots of other polymer fillings, which results in a higher contamination level of the recyclates. Well-controlled deposit recollection systems provided much lower contaminated recyclates [68]. For PS, migration appears to be high when in contact with fatty foods and when the surface/volume ratios are high [69]. In the case of mixed plastic packaging waste, the used packaging may come into contact with and interact with non-food substances, such as flavoring components and/or harmful contaminants [52]. Therefore, the substances found in polymers can differ greatly depending on the type of collection system [69]. Contaminants from non-food packaging materials are particularly noteworthy, as they might add substances to food packaging streams that are otherwise not permitted by the EFSA due to safety concerns. This makes a reliable recycling cycle considerably more difficult. In order to ensure safety standards during the recycling process, certain knowledge about the percentage of non-food polymers is therefore necessary. Franz & Welle [49] found that for the recycling process of PET recyclates, a share of 20% of non-food recyclates is considered to be safe. EFSA, however, limited the amount of non-food bottles in the input stream of PET recycling processes to 5% [12]. Since different countries have different collection systems and recyclates are traded on an international market, information on the first use and origin of the collected polymers is mostly not available. All of this shows the relevance of a standardized collection system. Current systems such as the curbside collection system, e.g., the green dot system, which was pioneered in Germany but has now been adopted by most European countries as well as Israel and Turkey, collects food and non-food packaging together, whereby cross-contamination between food and non-food packaging can occur [70]. For example, limonene from cosmetic packaging could be introduced into post-consumer recyclates in this way. In cosmetic products, it is used as an ingredient in the perfume composition [71]. An alternative could be so-called deposit systems, where the packaging is separated into food and non-food packaging [14].

The importance of a thorough recycling system can also be seen when looking at the different chemical properties of the virgin base polymer and post-consumer material substances. As we could show, virgin base polymer substances were significantly less often classified as toxic than post-consumer substances in regard to Cramer class > I and ngtc alerts. Therefore, the sorption of post-consumer contaminants and other NIAS into the polymer matrix must be kept as low as possible.

In addition to a more efficient collection system of food packaging, the cleaning process of polymers is also important. New technological innovations such as the super clean flake process can achieve high cleaning efficiencies and thus ensure a higher quality of post-consumer recyclates [14]. In PET, a cleaning efficiency of super clean processes of 90% for all contaminations is considered as a worst-case. Cleaning efficiencies between

90% and 99% are common, depending on the material [49]. Here, too, the efficiency of the process depends, among other things, on the molecular weight of the substances. It is worth mentioning that low molecular contaminants are more easily absorbed but are thus also more easily removed by the cleaning process [48,49,53,72,73].

Polarity also has an influence on the sorption behavior of contaminants and NIAS in plastic packaging. Since it has often been described in the literature that non-polar substances have a high affinity to plastics [29], it can be assumed that more non-polar than polar substances are found in polymers. According to Endo & Koelsmans [74], however, a high log $P$ value is not an indication of a strong interaction between the substance and the polymer. The log $P$ value can thus only serve as an initial estimate of the possible diffusion behavior [75]. This is because the partitioning coefficient only measures the distribution equilibrium, while the diffusion coefficient measures at which rate the substances are absorbed, as well as the distribution equilibrium [76].

We could show a significant correlation between the non-polar substances from log $P$ class 4 and the Cramer class > I and ngtc alerts. According to this, the proportion of toxicologically concerning substances, as described by Cramer, is higher for non-polar substances than for polar compounds. Since a correlation between non-polar substances and substances with gtc alerts could not be significantly demonstrated, a general correlation between the polarity and toxicological classification of substances was only partially verified.

### 3.4. Critical Evaluation of the Methodology and Outlook

Comparing the three Cramer classes in relation to toxicological endpoints such as the No Observed Effect Concentration (NOEC) (lowest concentration of a toxic substance at which no significant effect is induced in a test organism) shows considerable overlap in the distributions of NOEC values, which means that Cramer classes do not discriminate well between substances of different toxic potency [38]. Furthermore, factors such as bioaccumulation and metabolite/degradation are currently only considered to a small extent [34,38].

Since all unknown substances must be evaluated as toxicologically concerning, the proportion of unknown substances is important in the evaluation process of the polymers. The results of the polymers could therefore have been different if all substances had been identified. The results showed different proportions of unknown substances in the polymers. It is, therefore, important to take this into account when looking at the data.

In addition, there is a proportion of substances that are not detectable with state-of-the-art analytical methods applied for the evaluation of post-consumer recyclates. More sensitive methods might be available, but such methods need more information about the nature and concentration of the target substances compared to less sensitive screening methods. Such additional information is currently not available. Therefore, the presence of (geno)toxic classified substances can currently not be determined with certainty, resulting in a health safety risk. Further, the use of multiple analytical methods might help the detection of more substances. For example, it is possible that substances, such as non-volatile substances, were not detected by the here-used GC that could have been better detectable with HPLC. If the presence of DNA-reactive carcinogens or other critical substances could be excluded with certainty, the migration limit would increase by a factor of 600 and would mean an enormous breakthrough for the recycling of food packaging [77]. However, reliable detection of the broad spectrum of DNA-reactive carcinogens is considered impossible with classical analytical methods. A promising approach is in vitro bioassays. This method specifically detects DNA-damaging agents, including currently unknown carcinogens. Further, they show the biological effects of a combination of substances [78]. However, while this approach has been used with promising results for a broader application of in vitro bioassays to assess the safety of food packaging, some remaining challenges have to be solved [78–81]. This includes too high limits of detection (LOD) and the standardization and optimization of sample preparation processes [79,81].

For a successful implementation of other polymers, such as polyolefins or PS, in a circular economy, further research gaps must be closed, and new innovations must be developed. Due to the high diffusivity of polyolefins, it is difficult to imagine reliable safety standards in the future, whereas the properties of PS are more likely to suit the regulation of the EFSA for a safe circular economy. Thus, better analytical methods, as well as standardized recycling systems and cleaning processes, are needed.

## 4. Materials and Methods

### 4.1. Sample Description and Analysis

#### 4.1.1. Sample Collection

A total of 179 samples were analyzed, which can be categorized into the materials HDPE (50 samples), LDPE (27 samples), PE (9 samples), PET (25 samples), PP (40 samples) and PS (28 samples). All samples were post-consumer recyclate samples previously used for food and non-food applications and were obtained as commercially available post-consumer recyclates from cooperating companies which are not mentioned by name here.

#### 4.1.2. Sample Preparation and Analysis

Gas chromatography screening analyses were performed with both liquid extraction and headspace. In addition, target additives were quantified.

The qualitative screening analysis was done using liquid extraction with GC-FID and GC-MS detection. To this end, 1.0 g of each sample (duplicate) was extracted by total immersion with 10 mL dichloromethane (DCM) for 3 days at 40 °C (polyolefins, PET) and 10 mL acetone for 3 days at 60 °C (PS). An aliquot of the extracts was mixed with 50 μL of the internal standard solution tert-butyl-hydroxy anisole (BHA) and Tinuvin 234 (1000 ppm each) and analyzed for extracted components by GC-FID screening. Gas chromatographic separation of the extracts was done with a DB-1 capillary separation column, dimensions: 30 m length-0.25 mm inner diameter (i.d.)-0.25 μm film thickness. The temperature program was 50 °C (2 min isothermal) to 340 °C with a heating rate of 10 °C/min, then 10 min isothermal at 340 °C. The GC-MS system used was a ThermoFinnigan SSQ (Bremen, Germany) equipped with a DB-1-MS column, dimensions: 30 m length-0.25 mm i.d.-0.25 μm film thickness. The temperature program was 80 °C (2 min) with a heating rate of 10 °C/min, then 340 °C (30 min). The electrical ionization was done in full scan mode with a mass range of $m/z$ 40–800. This method was used to detect semi-volatile organic components in a molecular weight range from 80 to 700 Daltons.

For the headspace screening, 1.0 g of the samples was analyzed using headspace gas chromatography with FID. The headspace equilibration temperatures were used close to the melting point for polyolefins and the highest possible for the applied equipment for PET and were validated during hundreds of analyses. The gas chromatograph used was a Perkin Elmer AutoSystem XL (Rodgau, Germany) with a DB 1 column, dimensions: 30 m length-0.25 mm i.d.-0.25 μm film thickness. The temperature program was 50 °C (4 min) with a rate of 20 °C/min, then 320 °C (15 min). The pressure was 50 kPa helium with a split of 10 mL/min. The used headspace autosampler was a Perkin Elmer HS 40 XL. The oven temperature was 120 °C for polyolefins, 150 °C for polystyrene, and 200 °C for PET. The needle temperature was 210 °C, and the transfer line was 210 °C. The equilibration time was 1 h, the pressure build-up time was 3 min, the injection time was 0.02 min, and the withdrawal time was 1 min. Identification of the volatile compounds was made by a coupling of headspace GC with MS. The system used was a Perkin Elmer Clarus 600 HT-HS-GC-MS-System with a ZB 1 MS guardian column, dimensions: 30 m length-0.25 mm i.d.-0.25 μm film thickness. The temperature program was 40 °C (2 min) with a heating rate of 10 °C/min, then 320 °C (5 min). The headspace autosampler had an oven temperature of 120 °C for polyolefins, 150 °C for polystyrene, and 200 °C for PET. The needle- and transfer line temperature was 210 °C, the equilibration time was 1 h, the pressure build-up time was 3 min, the injection time was 0.04 min, and the withdrawal

time was 1 min. The electrical ionization was done in full scan mode with a mass range of *m/z* 35–300.

The quantification of the additives was done by external calibration using the respective fragment masses of the different additives. The solvent extracts of the determination of non-volatile components (see Section 4.1.2) were evaporated to dryness and redissolved in 95% ethanol. The ethanol solutions were analyzed by HPLC/MS in the full scan mode from 500 to 1200 Da. The mass spectrometer used was a Thermo LTQ (Dreieich, Germany) with a Waters Xbridge ODS 2 column, dimensions: 150 mm length-3 mm i.d.-particle size 5 μm with a gradient water/methanol.

### 4.2. Qualitative Evaluation of Chemicals in Post-Consumer Recyclates

The identification of the obtained GC-MS spectra was performed by comparison with the NIST spectral library (NIST/EPA/NIH Mass Spectral library, version 2.0 d:2005-04, MS Search 2.0) (Gaithersburg, MD, USA). This software provides reference mass spectra for GC-MS. At a value of 800 matches or higher, an identification was considered certain. In literature, a combination of GC-MS and GC-FID is usually run independently; then, afterward, the chromatograms are manually aligned, and the peaks are compared according to the retention time [82]. Therefore, to draw conclusions from the identified substances in the GC-MS samples to the samples measured with GC-FID, both chromatograms (MS and FID) of the same sample were compared by their peak retention times. Since the retention times of the same substance differ for both methods, the known retention times of the internal standards were used as points of reference. This manual peak-by-peak comparison was made for all GC-MS samples. Once these identified substances were assigned to their GC-FID retention times, these FID spectra were also compared peak-by-peak with the remaining FID samples. It was now possible to identify the substances in those samples that had only been measured with GC-FID. It has to be mentioned that FID-to-FID comparisons were only made within the same group of polymers.

In order to make assumptions about which substances have entered the polymers through the recycling process, the identified substances were classified into virgin base polymer substances (degradation product from the polymer or an additive) and alleged post-consumer substances (contamination). For unknown substances and identified substances to which no CAS number and/or SMILES code could be found, a worst-case scenario with the potentially highest toxicity was assumed.

### 4.3. Classification of Chemicals

#### 4.3.1. In Silico Characterization: Cramer Classification and (Non)Genotoxic Carcinogenicity Alert

The Cramer classification of the identified substances was performed by Toxtree (Estimation of Toxic Hazard-A Decision Tree Approach; Version 3.1.0-1851-1525442531402) (Sofia, Bulgaria). The Cramer scheme is a widely used approach for classifying and raking chemicals according to their expected level of oral systemic toxicity. The decision tree categorizes chemicals, mainly on the basis of chemical structure and reactivity, into three classes indicating a high (class III), medium (class II) or low (class I) level of toxicological concern [37,38].

Class I are substances with simple chemical structures and for which efficient modes of metabolism exist, suggesting a low order of oral toxicity. Class II are substances that possess structures that are less innocuous than class I substances but do not contain structural features suggestive of toxicity such as those substances in class III. Class III are substances with chemical structures that permit no strong initial presumption of safety or may even suggest significant toxicity or have reactive functional groups.

Alerts for genotoxic (gtc) or non-genotoxic carcinogens (ngtc) were determined using the integrated Toxtree decision tree "Carcinogenicity (genotox and non-genotox) and mutagenicity rule base by ISS model". From the point of view of the mechanism of action, carcinogens are classified into: (a) genotoxic carcinogens, which cause damage directly

to DNA, (b) epigenetic carcinogens that do not bind covalently to DNA, do not directly cause DNA damage, and are usually negative in the standard mutagenicity assays. While epigenetic carcinogens act through a large variety of different and specific mechanisms, genotoxic carcinogens have the unifying feature that they are either electrophiles or can be activated to electrophilic reactive intermediates [83]. The ISS models provided the endpoints 'carcinogen' or 'non-carcinogen'.

To compare the polymers among each other, the number of substances classified higher than Cramer class I was counted and evaluated, as well as the number of substances with potential (non)genotoxic carcinogenicity alerts.

### 4.3.2. Determination of Log *P* Values

The log *P* values of the identified substances were calculated using the free online software Mol Inspiration (miLogP2.2-November 2005). To ensure better comparability, all log *P* values were divided into four classes: class 1: log *P* = 0–1; class 2: log *P* = 1–2; class 3: log *P* = 2–4; class 4: log *P* $\geq$ 4. It should be mentioned that none of the detected substances had an estimated log *P* value lower than 0.

### *4.4. Data Evaluation*
### 4.4.1. Calculation of Diversity Indices

Ecological indices were used to determine the chemical diversity of each polymer. Therefore, each polymer was treated as a different "habitat" while each substance was treated as a "species". To determine the number of an identified substance, it was counted in how many samples of the same polymer this substance was found. Resulting from this, the chemical diversity of each polymer type was determined using the Shannon index, Simpson index, and evenness. Both indices predict the chances of finding an individual of a species that is randomly taken from a data set [41,42]. Evenness is a measure of unequal distribution and indicates how frequently individuals of a species occur in a certain habitat in relation to the number of individuals of other species occurring in the same habitat [43].

### 4.4.2. Statistical Analysis

Analyses of variance (ANOVAs) were applied to determine significant differences in the number of polymer-associated substances, their Cramer classification, and their polarity (Log *P*) between the polymers. Furthermore, ANOVAs were used to evaluate differences in Irgafos and Irganox concentrations between polymers. For post-hoc comparisons of each polymer with PET, Dunnett's-tests were performed. ANOVAs were checked for normality and homoscedasticity of residuals using quantile-quantile and residual vs. fitted plots. In case those assumptions were not met, the data were transformed by applying the decadic logarithm.

Polymer-specific differences in gtc and ngtc alerts of polymer-associated substances as well as their origin (post-consumer substances or virgin base polymer substances), were assessed using $X^2$-tests. Differences in the substance diversity (Shannon and Simpson indices, evenness) were estimated by testing their distribution against a uniform distribution using Kolmogorov-Smirnov (KS)-tests. Multiple comparisons on the same data were adjusted in *p* values using Bonferroni correction.

The significance level for all statistical tests was set at 0.05. Throughout the present study, the term significant is exclusively used with reference to statistical significance. R (Version 4.2.1) (Boston, MA, USA) was used for the statistical evaluation and the creation of figures.

## 5. Conclusions

It was shown that the investigated polymers differ both in the total number of identified substances and the number of toxicologically concerning substances. It was confirmed that PET contains fewer toxic classified substances and, in general, less NIAS than other polymers. This is due to the higher diffusivity of polyolefins. Accordingly, contaminants are more easily absorbed and adsorbed (in)to polyolefins and are, therefore, more difficult

to predict. The high diffusivity of polymers such as polyolefins thus challenges their use in a circular economy for food packaging in the near future. In this context, log *P* values were found to provide an approximate orientation but are not ultimately sufficient in determining the migration behavior of contaminants.

It was also shown that diffusivity is not the only determining factor for well-decontaminated post-consumer recyclates. This is because less diffusive PS contains a high total number of substances. Furthermore, collection systems, as well as cleaning processes, may considerably impact the reusability of post-consumer recyclates since it was shown that post-consumer substances were classified significantly more often as toxic than virgin base polymer substances. Additionally, it was shown that the most frequently found substances were either virgin base polymer substances (additives and degradation products) or post-consumer substances, out of which a few substances were classified as Cramer class III substances and/or had (n)gtc alerts and therefore might pose a potential health risk.

In order to fulfill the objective of a safe and reliable closed-loop circular economy, further research gaps must be closed, and new innovations must be developed. Moreover, better detection methods are needed to ensure that all substances in a polymer can be identified and classified on a routine basis. This would allow the identification of substances that cannot be detected with current detection methods and would greatly improve the assurance of the safety of recycled food-packaging. In addition to the Cramer classification applied here, in vitro bioassays thus appear to be an essential extension of this hitherto purely theoretical approach.

**Author Contributions:** Conceptualization, K.M., C.R. and F.W.; methodology, A.G. and K.M.; software, A.G., C.R., A.S. and Z.S.; validation, K.M., C.R., A.S. and F.W.; data curation, A.G., C.R. and A.S.; writing—original draft preparation, C.R.; writing—review and editing, K.M., C.R., A.S., Z.S. and F.W.; supervision, F.W.; project administration, F.W.; funding acquisition, F.W. All authors have read and agreed to the published version of the manuscript.

**Funding:** The IGF project 258 EN of the research association Industrievereinigung für Lebensmittel-technologie und Verpackung e. V.-IVLV, Giggenhauser Str. 35, 85354 Freising, Germany, was funded by the German Federal Ministry of Economics and Climate Protection through the German Federation of Industrial Research Associations (AiF) within the framework of the program for the promotion of joint industrial research and development (IGF) on the basis of a resolution of the German Bundestag.

**Data Availability Statement:** The data presented in this study are available on request from the corresponding author.

**Acknowledgments:** Thanks are due to Christian Kirchnawy, Elisa Mayrhofer, Ida Peneder, Elisabeth Pinter, Michael Washüttl, Silvia Apprich, Rainer Bernhard, Elisabeth Haider, Eva Maria Ortner, Manfred Tacker, Verena Vogler, Ay Zafer, Martin Ramsl, Michael Barwitz, Kristina Böck, Silvia Demiani, Mladen Juric, Alexandra Mauer, Katarina Willer, Gabriele Gedik, Anita Magstl and Tobias Voigt. Further thanks are due to all cooperating companies that were involved in this research project.

**Conflicts of Interest:** The authors declare no conflict of interest. The funders had no role in the design of the study; in the collection, analyses, or interpretation of data; in the writing of the manuscript; or in the decision to publish the results.

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
