# Peer review of "Identification and Evaluation of (Non-)Intentionally Added Substances in Post-Consumer Recyclates and Their Toxicological Classification"

_recycling, doi:10.3390/recycling8010024_

Round 1

Reviewer 1 Report

The authors discuss an important topic with respect to utilizing post consumer materials for direct food contact. This manuscript presents critical data to regulators and brand owners regarding the potential to use recovered polymers and the potential health and environmental risks. A few items need to be addressed prior to publication.

General comments:

Where any of the samples analyzed post-industrial or were all post-consumer? What were the original applications- food vs non food?

How as the HS temperatures selected? some polymers would be close to melt (LDPE) and others not (PET).

Why do the authors think that LDPE had the lowest number of substances? this seems counter intuitive for a polymer with a higher diffusion coefficient than PET. Describing the original application of these polymers by classification of application would be useful.

Specific comments by line number

18 Mention the specific limit set by EFSA, please.

61,62 what about other IAS components? For example, plasticizers and adhesives?

117 Please provide a reference for this section.

Figure 1: Need to make it more clear in describing the boxplot. Is the SD coming from the sample replicate? Please clarify the sample replicates number.

149 The largest proportion of the total substances was found in HDPE at what percentage? As you mentioned In PP in percentage.

158 Place reference for ‘diversity indices (Shannon and Simpson)’.

231 Do you mean that there was no NIAS in the sample? How can the authors confirm without identification of the original application?

258 What do you mean by example substance?

341 Do you have a result that which content was higher in your sample NIAS or IAS (intentionally added substances)?

352 I think this information has been already mentioned before.

352 Is Limonene food grade additive? If so, is there any concentration limit for use? At what conc. did you find it? Mention please.

375 What concentration is considered higher for Limonene?

382 Can you please be a little specific that what you want to emphasize by saying the type of collection system and what effect you see from that in your experiment?

401 Please mention which country use ‘Green dot system’

417 can you please add the satisfying cleaning efficiencies percentage for each material.

420 did you intentionally use adsorbed twice?

454 Discuss state-of-the-art analytical methods.

518 Add reference for Fraunhofer IVV inhouse method.

Overall, a good impactful scientific article that is anticipated to be cited often.

Author Response

Dear Reviewer, Thank you for the helpful comments. You will find the answers in the attached document. 

Reviewer 2 Report

The authors describe a method to classify post-consumer waste plastic according to their potential toxicity in food packaging applications. The topic is definitely interesting for the recycling and packaging community, however, major revisions are necessary before accepting the paper for publication.

Major revisions.

Point 1: the title and the nomenclature are confusing throughout the paper. Terms such as “post-consumer recyclates” (title), “recycled polymers” (line 114), “post-consumer samples” (line 120) are apparently used as synonyms, although they are not. It would be appropriate to define in the introduction what the authors mean with recycled polymers and post-consumer samples, and be consistent throughout the text. I would consider post-consumer recycled polymers, for instance, those materials that have been used, disposed, collected, sorted and re-processed. To my opinion, post-consumer indicates materials that were used and disposed.

Point 2: the structure of the paper is confusing. The materials and methods are described after the results and discussion and before the conclusions. The reader has to jump back and forth to find all the needed information to understand the content.

Point 3 (line 169): the use of the definition virgin material is misleading. My opinion is that it would be better to use base polymer instead or a similar definition. The materials analysed are post-consumer recyclates, which means that they are a blend of many different virgin polymers, additives, degradation products and possible contaminations from other polymers. The chromatographic analysis showed presence of fragments of the base polymer not of the virgin material because for a post-consumer recycled material it is not possible to know the original virgin polymer, including its stabilization recipe. Additionally, in which table or charts are the results described? A cross-reference or charts/tables are missing.

Point 4 (lines 224-264): the actual results of the qualitative substance identification are only partially described. It would be beneficial to broaden Table 3 with the 20-30 substances that were most frequently detected, including mass spectra match quality as well. Furthermore, it would be interesting to see the detected mass spectra and their corresponding references for the substances that were quantified. It is not necessary to add the information into the results section, but it could be included at least in the supplementary information.

Point 5: the sample collection chapter is not properly described. I might understand the reason for non disclosing the suppliers names, however it would be beneficial to give some more information about how those materials were collected and identified. More specifically: A) Are the samples post-consumer recyclates or post-consumer waste samples? B) Are the samples coming from sorting plant or recyclers, purchased as commercially available post-consumer recyclates or a mix of the above? C) What about the possible contaminations from other polymers into the materials analysed? It is common, for instance, that PP recyclates have a certain PE content and vice versa. How would this influence the results? D) How the authors did identify the polymers? Did they use FTIR ATR spectroscopy, other analytical methods or did they use information from datasheets/suppliers?

Author Response

(The authors gave the same response as above.)

Round 2

Reviewer 2 Report

Dear authors, thank you for having considered the comments and modified the manuscript accordingly. There is only one point left, from my side. 

Table 3: The new version of the table significantly improved compared to the previous one. However, I would suggest to double-check the last column (origin of the substance detected) and to provide references for the assignments, as some of them might be not correct. For instance, Di-(ethylhexyl)-isophthalat (390) - 137-89-3 is reported as PCM (from packed content). I am not so sure of it because DEHP is a typical plasticizer (additive) and it might be included in the polymer formulation itself or coming from contaminations with other polymers (such as PVC) during recycling. Furthermore, I would suggest to give more information regarding the possible origin, such as  detergent, cosmetic, fat residue and so on (as examples). 

Author Response

You can find the response in the document attached.
